# Floquet Fractional Chern Insulators and Competing Phases in Twisted Bilayer Graphene

Peng-Sheng Hu, Zhao Liu⋆

Zhejiang Institute of Modern Physics, Zhejiang University, Hangzhou 310027, China
⋆ zhaol@zju.edu.cn

July 19, 2022

## Abstract

We study the many-body physics in twisted bilayer graphene coupled to periodic driving of a circularly polarized light when electron-electron interactions are taken into account. In the limit of high driving frequency $\Omega$, we use Floquet theory to formulate the system by an effective static Hamiltonian truncated to the order of $\Omega^{-2}$, which consists of a single-electron part and the screened Coulomb interaction. We numerically simulate this effective Hamiltonian by extensive exact diagonalization in the parameter space spanned by the twist angle and driving strength. Remarkably, in a wide region of the parameter space, we identify Floquet fractional Chern insulator states in the partially filled Floquet valence band. We characterize these topologically ordered states by ground-state degeneracy, spectral flow, and entanglement spectrum. In regions of the parameter space where fractional Chern insulator states are absent, we find topologically trivial charge density waves and interaction-induced Fermi liquid which strongly compete with fractional Chern insulator states.

# 1  Introduction

Van-der-Waals heterostructures with moiré patterns [1, 2] have attracted tremendous attention over the last few years. When two atomic layers are stacked with each other, the mismatch between two crystals due to different lattice constants and/or a twist angle generates a large-scale superlattice and affects the interlayer coupling. Such moiré systems, whose band structures are highly controllable, are promising to host numerous exotic phenomena. A representative example is the twisted bilayer graphene (TBG), consisting of two sheets of monolayer graphene with a twist angle in between [3]. One of the most striking features of TBG is that, at certain special twist angles (called magic angles), the low-energy bands near charge neutrally point (CNP) can be tuned to be flat, thus providing an ideal platform to investigate correlated physics. Indeed, unconventional superconductivity and ferromagnetism have been discovered in TBG [4–9]. Another salient progress in this direction is the observation of the quantum anomalous Hall effect at zero magnetic field (also called Chern insulators) in TBG aligned with a hexagonal boron nitride (hBN) [10]. In this case, hBN gaps out the protected Dirac points of TBG, such that the flat bands around CNP is isolated and acquire nonzero Chern numbers [6, 10–14]. Motivated by this topological band structure, a lot theoretical efforts have been made to demonstrate the possibility of realizing the zero-field fractional Chern insulators (FCIs) [15–17] in TBG-hBN when the flat bands near CNP are partially filled by interacting electrons [18–21]. Excitingly, evidence of FCIs in TBG-hBN has been reported in a recent experiment at weak magnetic fields [22].

External driving fields of light provide an alternative avenue to obtain topological band structures even if the original bands in the absence of driving is topologically trivial [23–30]. Unlike nondriven systems, systems periodically driven by light do not have well-defined static band structures. However, the effect of light enters an effective static Hamiltonian that captures the dynamics of the system on time scales much longer than the driving period [24, 31–33]. This effective static Hamiltonian, which describes photo-dressed band structures, has been extensively used to predict the out-of-equilibrium topological properties of various periodically driven systems [23, 24, 27, 28, 34]. In particular, accompanying the rapid development of moiré materials, exploring the effects of light on moiré band structures has become an intriguing direction recently [35–44]. Just like in monolayer graphene [23, 24, 45], a circularly polarized light can open a band gap at the Dirac points of TBG and give rise to Floquet topological flat bands [36, 37, 39]. In this sense, for TBG the light driving plays the role of hBN in the static case.

While exciting progress has been made on Floquet moiré materials, the interactions between electrons are not yet taken into account in previous works [36, 37, 39–44]. Hence whether correlated topological phases can be induced by light driving in these systems remains a crucial open question. In the context of photon-dressed topological band structure, a natural candidate of correlated topological states is the Floquet FCI which was first proposed in monolayer graphene [46,47]. The Floquet FCI is formed when interacting particles partially fill in a topological flat band of the effective static Hamiltonian of the driven system.

In this work, to investigate the possibility of stabilizing Floquet FCIs in moiré materials, we consider TBG driven by a circularly polarized laser light. We focus on the limit of high driving frequency $\Omega$, in which case we truncate the effective static Hamiltonian to the order of $\Omega^{-2}$. Unlike Ref. [47], we find this effective static Hamiltonian does not contain new effective interactions that could diminish FCIs. This feature is due to the special form of the TBG Hamiltonian. Then we use exact diagonalization to search for the evidence of Floquet FCIs in the parameter space spanned by the twist angle and the driving strength. Strikingly, for electrons interacting via the screened Coulomb potential, we find a wide region in which FCIs are promising to exist when the Floquet valence band is occupied at filling $\nu = 1/3$. We

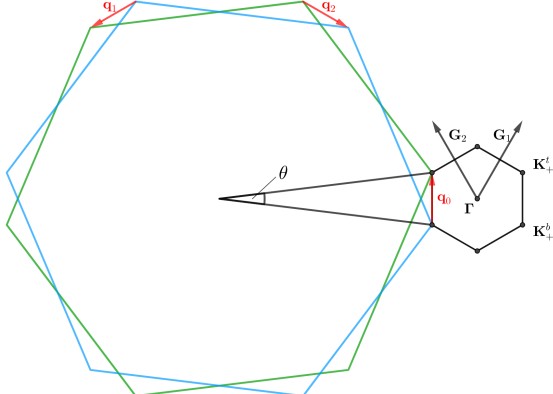

Figure 1: Moiré Brillouin zone of TBG. Two large hexagons are original first Brillouin zones of top and bottom graphene layers. The small hexagon represents the MBZ resulting from twist. The $\mathbf{K}_+^{t,b}$ points, the vectors $\mathbf{q}_{0,1,2}$ in Eq. (1) and the MBZ reciprocal lattice vectors $\mathbf{G}_1$ and $\mathbf{G}_2$ are given.

also observe charge density waves (CDWs) and Fermi liquid (FL)-like states in the neighboring regions of FCIs. The overall phase diagram is similar for different interlayer coupling strength of TBG. Because the effective Hamiltonian of Floquet TBG in the high-frequency limit is analogous to that of static TBG-hBN, our results could also apply to the many-body physics in the later system.

## 2 Model

We consider the low-energy dynamics of TBG at small twist angles ($\theta \sim 1°$) by following Bistritzer and MacDonald's continuum model [48]. After twisting, the original first Brillouin zone of single-layer graphene is folded into moiré Brillouin zones (MBZ). We assume valley and spin polarization and focus on the MBZ near the valley $\mathbf{K}_+$ of the single-layer graphene (Fig. 1). In this MBZ, the $\mathbf{K}_+$ points of the top and bottom graphene layers are located at $\mathbf{K}_+^t = R_{\theta/2}\mathbf{K}_+$ and $\mathbf{K}_+^b = R_{-\theta/2}\mathbf{K}_+$, respectively, with $R_\theta$ a counter-clockwise rotation around the $z$-axis in the momentum space. $\mathbf{G}_1$ and $\mathbf{G}_2$ are the primitive reciprocal lattice vectors of the MBZ, with $\mathbf{a}_1$ and $\mathbf{a}_2$ the corresponding real-space primitive lattice vectors.

### 2.1 Static system

In the absence of driving, the single-electron Hamiltonian of TBG in the momentum space [18, 49] takes the form of

$$
\begin{aligned}
H_{\text{kin}} &= \sum_{\mathbf{k}} \left( \psi_t^\dagger(\mathbf{k}) h_{-\theta/2}\left(\mathbf{k} - \mathbf{K}_+^t\right) \psi_t(\mathbf{k}) + \psi_b^\dagger(\mathbf{k}) h_{\theta/2}\left(\mathbf{k} - \mathbf{K}_+^b\right) \psi_b(\mathbf{k}) \right) \\
&+ \sum_{\mathbf{k}} \sum_{j=0}^{2} \left( \psi_t^\dagger\left(\mathbf{k} - \mathbf{q}_0 + \mathbf{q}_j\right) T_j \psi_b(\mathbf{k}) + \text{H.c.} \right),
\end{aligned} \tag{1}
$$

where $\psi_t(\mathbf{k}) = \begin{pmatrix} \psi_{tA}(\mathbf{k}) \\ \psi_{tB}(\mathbf{k}) \end{pmatrix}$ and $\psi_b(\mathbf{k}) = \begin{pmatrix} \psi_{bA}(\mathbf{k}) \\ \psi_{bB}(\mathbf{k}) \end{pmatrix}$ are spinors of annihilation operators for electrons in top ($t$) and bottom ($b$) graphene layers, respectively, and $A$ and $B$ correspond to the two sublattices in single-layer graphene. The first two terms in $H_{\text{kin}}$ are the Dirac Hamiltonians of top and bottom graphene layers, for which $h_\theta(\mathbf{k}) = h(R_\theta \mathbf{k})$ with $h(\mathbf{k}) = -\hbar v_F(k_x \sigma_x + k_y \sigma_y)$.

Here $\hbar v_F = \sqrt{3}at_0/2$, with $t_0 = 2.62$ eV and $a = 0.246$ nm the nearest-neighbor hopping amplitude and lattice constant of single-layer graphene, respectively. The third term describes the moiré tunneling between the two graphene layers. Such tunneling is encoded in the matrix

$$T_j = w_0 - w_1 e^{i(2\pi/3)j\sigma_z}\sigma_x e^{-i(2\pi/3)j\sigma_z} \tag{2}$$

with the momenta $\mathbf{q}_0 = R_{-\theta/2}\mathbf{K}_+ - R_{\theta/2}\mathbf{K}_+$, $\mathbf{q}_1 = R_{2\pi/3}\mathbf{q}_0$ and $\mathbf{q}_2 = R_{-2\pi/3}\mathbf{q}_0$ (Fig. 1). $w_0$ and $w_1$ in $T_j$ are the tunneling strengths between $AA$ and $AB$ sites, respectively. *Ab initio* numerics gives $w_1 \approx 110$ meV [50], however, some works suggested smaller values [18, 21, 51, 52]. In this paper, we consider two situations with $w_1 = 90$ meV and $w_1 = 110$ meV to account for variations of different theoretical models and realistic samples. Furthermore, we fix $w_0 = 0.7w_1$ to include the effects of lattice relaxation [53, 54] and corrugation [52, 55, 56]. To calculate the band structure at momentum $\mathbf{k}_0$ in the MBZ, we write $\mathbf{k}$ in $H_{\text{kin}}$ as $\mathbf{k}_0 + m\mathbf{G}_1 + n\mathbf{G}_2$ with integers $m, n = -d, ..., d$ then diagonalize $H_{\text{kin}}$, where $d$ is a suitably chosen cutoff. As there is no alignment with the hexagonal boron nitride (hBN) substrate, the Dirac band touching exists at the corners of the MBZ.

We simulate the interaction between electrons via the screened Coulomb potential

$$H_{\text{int}} = \frac{1}{2}\sum_{\mathbf{q}} V(\mathbf{q}) : \rho(\mathbf{q})\rho(-\mathbf{q}) :, \tag{3}$$

where $\rho(\mathbf{q})$ is the valley and spin projected density operator and $::$ means the normal order. We choose the Yukawa potential $V(\mathbf{q}) = \frac{e^2}{4\pi\epsilon_r\epsilon_0 S}\frac{2\pi}{\sqrt{|\mathbf{q}|^2 + \kappa^2}}$ to describe the screening, where $e$ is the electron charge, $\epsilon_r$ is the relative dielectric constant of the material, $\epsilon_0$ is the dielectric constant of vacuum, $S$ is the area of the moiré superlattice, and $\kappa$ measures the screening strength. Throughout this work, we fix $\epsilon_r = 4$ [57, 58] and $\kappa = 1/a_M$, with $a_M = a/(2\sin(\theta/2))$ the lattice constant of TBG.

## 2.2 Floquet system

Now we consider the coupling of TBG with light in the scenario of periodic driving. We suppose the system is driven by a circularly polarized light which shines vertically and uniformly across the surface of TBG. The light field is represented by an electric field rotating in-plane as $\mathcal{E} = \mathcal{E}_0(\sin\Omega t, \cos\Omega t)$, where $\Omega$ is the driving frequency. The corresponding vector potential $\mathbf{A} = A_0(\cos\Omega t, -\sin\Omega t)$, satisfying $\mathcal{E} = -\frac{\partial\mathbf{A}}{\partial t}$. For the single-electron Hamiltonian, the light field only affects the intralayer hopping. This is because the interlayer tunneling is dominated by hopping between atoms that are exactly on top of each other, thus mostly contributed by $z$-component of the vector potential which is absent in our setup [36–38]. We include the effect of light using a Peierls substitution $\mathbf{k} \rightarrow \mathbf{k} + e\mathbf{A}(t)/\hbar$ in the intralayer terms of Eq. (1), resulting in a time-dependent single-particle Hamiltonian $H_{\text{kin}}(t)$. The interaction Hamiltonian Eq. (3) remains as in the static case since it is expressed in terms of the density operator [46, 59]. Combining both terms, we get a new time-periodic Hamiltonian

$$H(t) = H_{\text{kin}}(t) + H_{\text{int}}. \tag{4}$$

to describe our system irradiated by the circularly polarized light, where $H(t) = H(t + 2\pi/\Omega)$.

According to Floquet theory, the stroboscopic evolution of the system, upon a unitary transformation, can be captured by an effective static Hamiltonian $H_{\text{eff}}$ that does not depend on initial conditions [31–33]. While in general it is complicated to evaluate $H_{\text{eff}}$, we consider the limit where the driving frequency $\Omega$ is large compared to other characteristic energy scales in the system. In this case, $H_{\text{eff}}$ can be represented by a series expansion of $1/\Omega$ [31–33, 47, 59]:

$$H_{\text{eff}} = H_{\text{eff}}^{(0)} + H_{\text{eff}}^{(1)} + H_{\text{eff}}^{(2)} + \dots, \tag{5}$$

with

$$H_{\text{eff}}^{(0)} = H_0, \tag{6a}$$

$$H_{\text{eff}}^{(1)} = \frac{1}{\hbar\Omega}\sum_{m=1}^{\infty}\frac{[H_m, H_{-m}]}{m}, \tag{6b}$$

$$H_{\text{eff}}^{(2)} = \frac{1}{(\hbar\Omega)^2}\left\{\sum_{m=1}^{\infty}\frac{[H_m,[H_0,H_{-m}]]}{2m^2}\right.$$
$$+ \sum_{\substack{m,m'=1\\m\neq m'}}^{\infty}\frac{[H_{-m'},[H_{m'-m},H_m]]-[H_{m'},[H_{-m'-m},H_m]]}{3mm'}$$
$$\left.+\text{H.c.}\right\}. \tag{6c}$$

Here $H_m$ is the Fourier transform of $H(t)$, i.e., $H(t) = \sum_m H_m e^{im\Omega t}$. For our model, $H_m$ is nonzero only when $m = 0, \pm 1$.

The zeroth order term $H_{\text{eff}}^{(0)} = H_0$ is just the static Hamiltonian $H_{\text{kin}} + H_{\text{int}}$. The first order term is

$$H_{\text{eff}}^{(1)} = \frac{(eA_0 v_F)^2}{\hbar\Omega}\sum_{\mathbf{k}}\left(\psi_t^\dagger(\mathbf{k})\sigma_z\psi_t(\mathbf{k}) + \psi_b^\dagger(\mathbf{k})\sigma_z\psi_b(\mathbf{k})\right), \tag{7}$$

which is the same as that derived for non-interacting TBG [36, 37, 39, 60] because the interaction is time-independent in our model. $H_{\text{eff}}^{(1)}$ is a single-particle term which introduces a staggered potential of strength $P = (eA_0 v_F)^2/(\hbar\Omega) = (3a^2 t_0^2 e^2 \mathcal{E}_0^2)/(4\hbar^3\Omega^3)$ in both graphene layers. As this stagger potential breaks the $C_2$ sublattice symmetry, it gaps the Dirac cones of the static TBG. Then the two flat bands near the CNP may be isolated and carry non-zero Chern number in specific range of $P$ and twist angle $\theta$ [36, 39]. Note that $H_{\text{eff}}^{(1)}$ plays a similar role to the alignment with an hBN substrate, which, to the lowest order, also introduces a staggered potential of about 15 meV [14]. The advantage of light driving compared to hBN is that we can easily control the strength of this staggered potential by tuning the electric field $\mathcal{E}_0$ or frequency $\Omega$.

Previous works studying high-frequency driving in non-interacting lattice models often neglect $H_{\text{eff}}^{(2)}$ and other higher-order terms in $H_{\text{eff}}$, because their corrections to the single-particle Hamiltonian is quite small. However, once interactions are considered, one should be very careful when dealing with these high-order terms, because they can include effective interactions even though the original interaction $H_{\text{int}}$ is time-independent. To the leading order, these effective interactions are present in $H_{\text{eff}}^{(2)}$ if $[H_m,[H_{\text{int}},H_{-m}]] \neq 0$. While being much weaker than $H_{\text{eff}}^{(0)}$ and $H_{\text{eff}}^{(1)}$, these effective interactions may still be comparable to the many-body gap protecting the ground state of $H_{\text{eff}}$, thus having essential influences on the low-energy stroboscopic physics. Indeed, it was found in some Floquet topological lattice models that the effective interactions in $H_{\text{eff}}^{(2)}$ led by original density-density repulsions destabilize topologically ordered FCIs [47]. In our model, we carefully evaluate $[H_1,[H_{\text{int}},H_{-1}]]$. Remarkably, we find it is zero due to the special forms of our $H_{\pm 1}$ and $H_{\text{int}}$ (see Appendix). Therefore, by contrast to Ref. [47], $H_{\text{eff}}^{(2)}$ in our model is still a single-particle correction without effective interactions:

$$H_{\text{eff}}^{(2)} = \frac{(eA_0 v_F)^2}{(\hbar\Omega)^2}\left[-H_{\text{kin}} + \sum_{\mathbf{k}}\sum_j\left(\psi_t^\dagger(\mathbf{k}-\mathbf{q}_0+\mathbf{q}_j)\mathbb{W}_\theta\psi_b(\mathbf{k})+\text{H.c.}\right)\right], \tag{8}$$

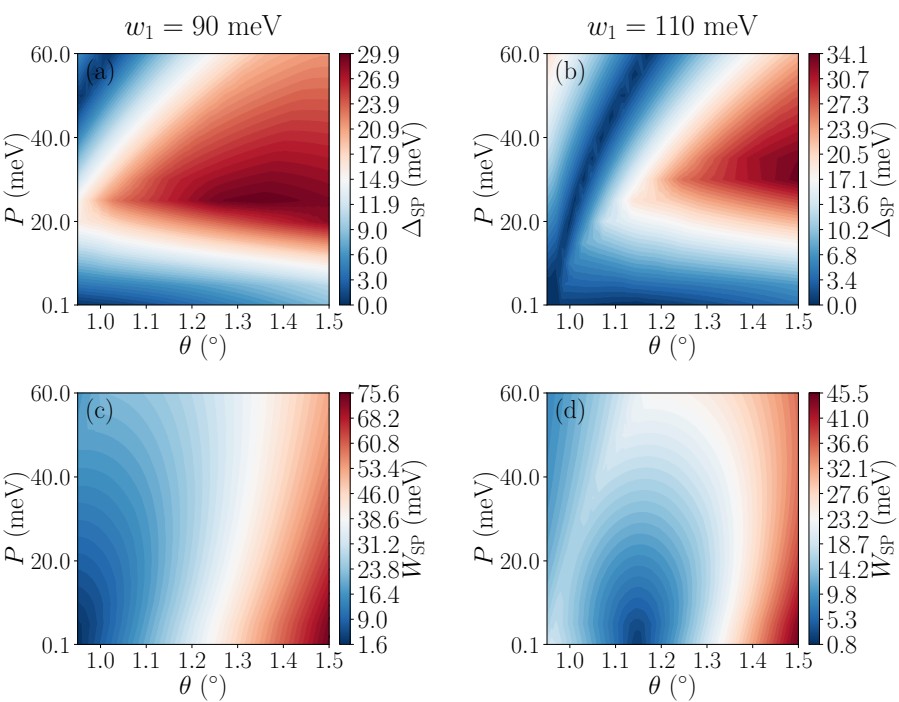

Figure 2: The indirect band gap $\Delta_{\text{SP}}$ and the bandwidth $W_{\text{SP}}$ of the Floquet valence band for $w_1 = 90$ meV [(a),(c)] and $w_1 = 110$ meV [(b),(d)].

where

$$\mathbb{W}_\theta = w_0 \begin{pmatrix} e^{i\theta} & 0 \\ 0 & e^{-i\theta} \end{pmatrix}. \tag{9}$$

In the following, we choose a high frequency $\hbar\Omega = 1.5$ eV, which significantly exceeds the energy scale of low-energy bands in static TBG. When $\Omega$ is fixed, $P$ quantifies the driving strength. We consider $P$ up to 60 meV, corresponding to a strong electric field $\mathcal{E}_0 \approx 8$ MV/cm. Then the prefactor $\frac{(eA_0 v_F)^2}{(\hbar\Omega)^2}$ of $H_{\text{eff}}^{(2)}$ is only $\sim 4\%$ at most. In this sense, our Floquet model could also simulate the static TBG aligned with hBN both on the top and at the bottom.

## 3 Floquet band structure

Now we have obtained the effective static Hamiltonian

$$H_{\text{eff}} = H_{\text{kin}} + H_{\text{int}} + H_{\text{eff}}^{(1)} + H_{\text{eff}}^{(2)} \tag{10}$$

describing the stroboscopic nature of our Floquet system, which includes the original interaction and a single-particle part corrected by driving. Before we dive into the interaction induced many-body physics, let us first analyze the properties of the Floquet bands. In the following, we focus on the valence band below the CNP.

In Fig. 2, we present the indirect band gap $\Delta_{\text{SP}}$ and bandwidth $W_{\text{SP}}$ of the valence band as functions of $\theta$ and $P$. For most parameters that we consider, we find positive $\Delta_{\text{SP}}$, meaning that the Floquet valence band is isolated from other bands at these parameters. However, there are some lines in the $(\theta, P)$ space along which the band gap vanishes. After calculating the Chern number $C$ of the Floquet valance band, we find these gap-vanishing lines correspond to the transition between $C = -1$ and $C = 0$ (Fig. 3). In the parameter range that we consider, the largest band gaps appear in the topological $C = -1$ case.

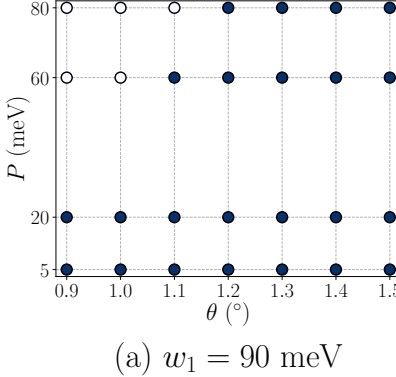
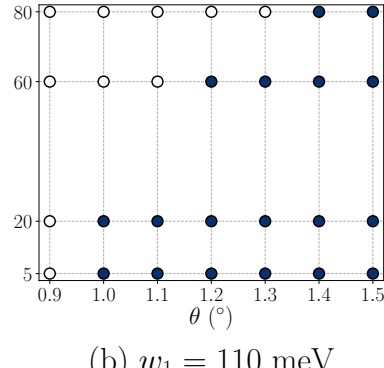

(a) $w_1 = 90$ meV                                      (b) $w_1 = 110$ meV

Figure 3: Chern number of the Floquet valence band for (a) $w_1 = 90$ meV and (b) $w_1 = 110$ meV. Solid dots represent $C = -1$, and circles represent $C = 0$.

## 4   Many-body physics

Because the Floquet valence band is well isolated and carries Chern number $C = -1$ in a wide range of parameters $\theta$ and $P$ (Figs. 2 and 3), it is promising to host Floquet FCIs. Now we consider the situation in which this band is partially filled by $N$ interacting electrons to confirm this possibility. We choose a finite periodic system with $N_1$ and $N_2$ moiré unit cells in the directions of the two primitive moiré lattice vectors, such that the band filling factor $\nu$ is defined as $N/(N_1 N_2)$. Due to the periodic boundary condition, each energy level of this finite system can be labeled by the total two-dimensional (2D) momentum $(K_1, K_2)$, with integers $K_1 = 0, \cdots, N_1 - 1$ and $K_2 = 0, \cdots, N_2 - 1$. Motivated by the observations of robust FCIs at $\nu = 1/3$ (lattice analogs of the celebrated Laughlin state [61]) in various static $|C| = 1$ topological flat bands [62, 63], we also fix $\nu = 1/3$ in our Floquet system.

We then focus on the $C = -1$ region in the $(\theta, P)$ space to numerically study the effective static Hamiltonian Eq. (10) to search for the evidence of FCIs. Since the Floquet valence band is well isolated in the $C = -1$ region, it is fair to project Eq. (10) to this active band, leading to

$$H_{\text{eff}}^{\text{proj}} = \sum_{\mathbf{k}} E(\mathbf{k}) c_{\mathbf{k}}^{\dagger} c_{\mathbf{k}} + \sum_{\{\mathbf{k}_i\}} V_{\mathbf{k}_1 \mathbf{k}_2 \mathbf{k}_3 \mathbf{k}_4} c_{\mathbf{k}_1}^{\dagger} c_{\mathbf{k}_2}^{\dagger} c_{\mathbf{k}_3} c_{\mathbf{k}_4}, \tag{11}$$

where $c_{\mathbf{k}}^{\dagger}$ ($c_{\mathbf{k}}$) is the operator creating (annihilating) an electron with momentum $\mathbf{k}$ in the Floquet valence band, $E(\mathbf{k})$ is the corresponding band dispersion, and the matrix element $V_{\mathbf{k}_1 \mathbf{k}_2 \mathbf{k}_3 \mathbf{k}_4}$ is given by [18]

$$V_{\mathbf{k}_1 \mathbf{k}_2 \mathbf{k}_3 \mathbf{k}_4} = \frac{1}{2} \delta'_{\mathbf{k}_1 + \mathbf{k}_2, \mathbf{k}_3 + \mathbf{k}_4} \sum_{\mathbf{G}} V(\mathbf{k}_1 - \mathbf{k}_4 + \mathbf{G}) \times \langle u(\mathbf{k}_1)|u(\mathbf{k}_4 - \mathbf{G})\rangle \langle u(\mathbf{k}_2)|u(\mathbf{k}_3 + \mathbf{G} + \delta\mathbf{G})\rangle. \tag{12}$$

Here $\delta'_{\mathbf{k},\mathbf{k}'}$ is the 2D periodic Kronecker delta function with period of MBZ reciprocal lattice vectors, $|u(\mathbf{k})\rangle$ is the Floquet valence band eigenvector, $\delta\mathbf{G} = \mathbf{k}_1 + \mathbf{k}_2 - \mathbf{k}_3 - \mathbf{k}_4$, and the sum of $\mathbf{G}$ is over the entire reciprocal space rather than only in a single MBZ. $E(\mathbf{k})$ and $|u(\mathbf{k})\rangle$ can be obtained by diagonalizing the single-particle part of $H_{\text{eff}}$. We then use exact diagonalization to extract the low-energy physics of this projected effective Hamiltonian Eq. (11) for $w_1 = 90$ meV and $w_1 = 110$ meV, respectively. We have examined that the ground-state energy gap obtained from diagonalizing Eq. (11) is always smaller than the single-electron band gap of the Floquet valence band, thus justifying the validity of band projection.

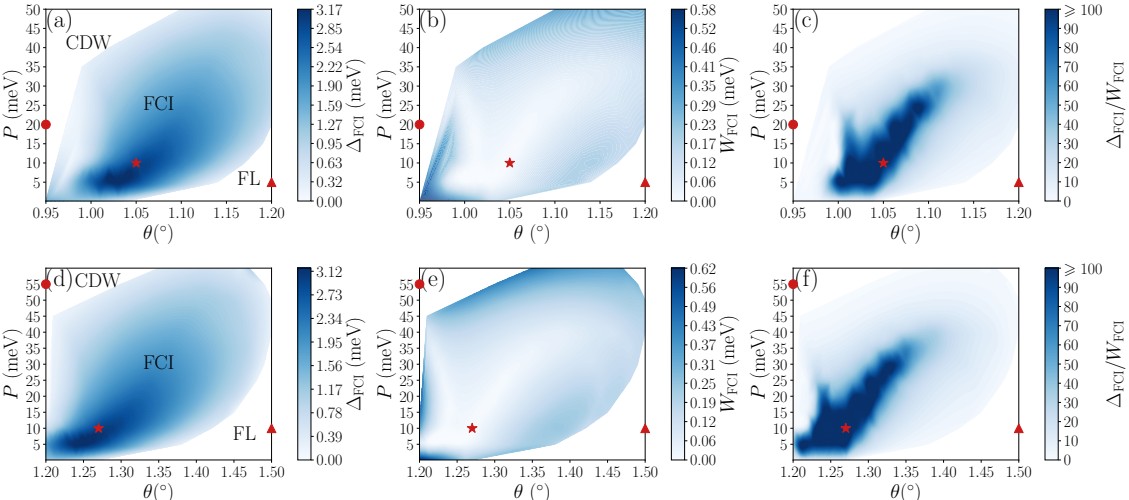

Figure 4: The FCI gap $\Delta_{\text{FCI}}$, FCI splitting $W_{\text{FCI}}$, and their ratio $\Delta_{\text{FCI}}/W_{\text{FCI}}$ for $N = 10$ and $N_1 \times N_2 = 5 \times 6$ with $w_1 = 90$ meV [(a)-(c)] and $w_1 = 110$ meV [(d)-(f)]. The three-fold FCI degeneracy is absent in white regions. In (a) and (d), we give the tentative phase diagram. As shown later, the CDW phase is identified by the structure factor, and the Fermi liquid (FL)-like phase is characterized by the step structure in the $n(\mathbf{k}) - E_h(\mathbf{k})$ curve. The markers indicate the representative parameter points that we choose in Figs. 5, 6, 7, 8 and 9.

## 4.1 $w_1 = 90$ meV

Let us first choose $w_1 = 90$ meV. On the torus geometry, an essential feature of the $\nu = 1/3$ FCIs is the robust three-fold ground-state degeneracy in momentum sectors determined by the Haldane statistics of the $\nu = 1/3$ Laughlin state [62, 64, 65]. Therefore, we compute the FCI gap $\Delta_{\text{FCI}}$ in the $C = -1$ region as the energy difference between the fourth and the first eigenvalues of the projected effective Hamiltonian Eq. (11), where all eigenvalues are sorted in ascending order. If the lowest three eigenstates are not in momentum sectors predicted by the Haldane statistics, we simply set the FCI gap to be zero. Meanwhile, we also measure the FCI splitting $W_{\text{FCI}}$, quantified by the energy difference between the third and the first eigenvalues when they are in the FCI momentum sectors. The result for $N = 10$ electrons is demonstrated in Figs. 4(a) and 4(b). Strikingly, we can identify a wide range of parameters for which the lowest three eigenstates are located in FCI momentum sectors, protected by a significant gap, and approximately degenerate [i.e., very large $\Delta_{\text{FCI}}/W_{\text{FCI}}$, as shown in Fig. 4(c)].

To examine the robustness of such ground-state degeneracies, we examine their dependence on the system size and the boundary condition. In Fig. 5(a), we demonstrate the low-energy spectra of the projected effective Hamiltonian for $N = 8$, 10, and 12 electrons at a representative parameter point $(\theta, P) = (1.05°, 10$ meV$)$ [labelled by the star in Figs. 4(a)-4(c)]. For each system size, we observe excellent three-fold ground-state degeneracy, i.e., the splitting of the three ground states are much smaller than their separation from higher-energy levels. A finite-size scaling of the ground-state splitting and the energy gap suggests that both the three-fold degeneracy and the ground-state gap are very likely to survive in the thermodynamic limit [Fig. 5(b)]. By inserting magnetic flux through the handles of the toroidal system, we find the three-fold ground-state degeneracy persists [Fig. 5(c)]. All these data further confirm the robustness of the three-fold topological degeneracy. Remarkably, the observed gap corresponds to a temperature of about 20 Kelvin, which is an order of magnitude higher than required by the conventional fractional quantum Hall states in two-dimensional electron gases.

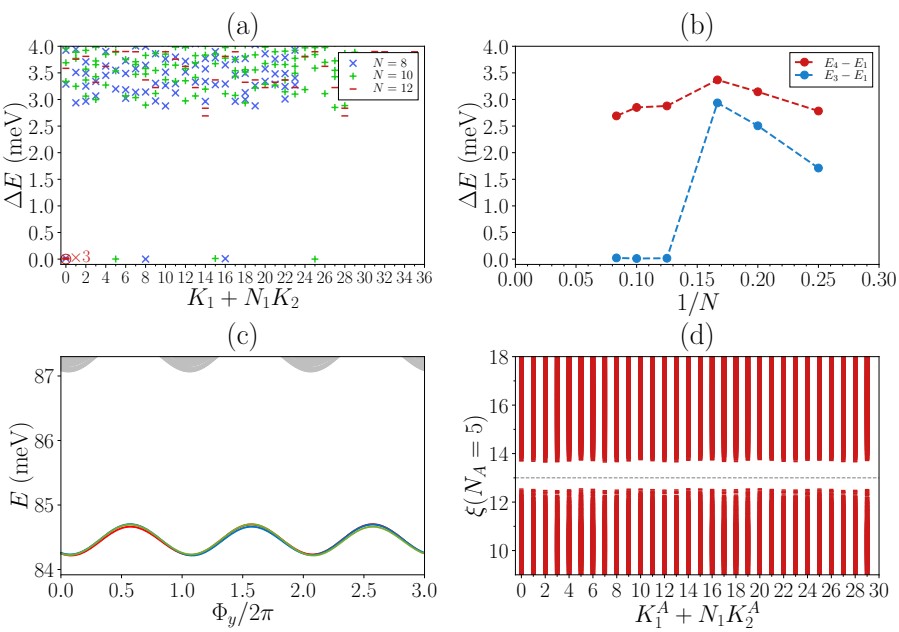

Figure 5: The $\nu = 1/3$ Laughlin FCIs in the Floquet valence band at $\theta = 1.05°$, $P = 10$ meV when $w_1 = 90$meV. (a) The low-lying energy spectra for $N = 8, 10, 12$ electrons on the $N/2 \times 6$ lattice. (b) The finite-size scaling of the energy gap $E_4 - E_1$ and the ground-state splitting $E_3 - E_1$ for $N = 4, 5, 6, 8, 10, 12$ electrons. (c) The spectral flow for $N = 10$, $N_1 \times N_2 = 5 \times 6$, where $\Phi_y$ is the magnetic flux insertion in the $\mathbf{a}_2$ direction. (d) The particle entanglement spectrum for $N = 10$, $N_1 \times N_2 = 5 \times 6$ and $N_A = 5$, with 23256 levels below the entanglement gap (the dashed line).

To further corroborate the nontrivial topological properties of the ground state, we also investigate the particle entanglement spectrum (PES), which encodes the information of quasi-hole excitations of the system and distinguishes FCIs from other competing phases [62, 66]. In Fig. 5(d), we divide the whole system into $N_A$ and $N - N_A$ electrons and label each PES level by the total momentum $(K_1^A, K_2^A)$ of those $N_A$ electrons. A clear entanglement gap appears separating the low-lying PES levels from higher ones, and the number of levels below the gap exactly matches the pertinent counting of quasihole excitations in the $\nu = 1/3$ Laughlin state [62, 64, 65]. This entanglement spectroscopy, together with the low-energy spectrum, strongly suggests that the most robust $\nu = 1/3$ Floquet FCIs exist in the region with $\theta \approx 1.0° - 1.1°$ and $P \approx 5$ meV $- 30$ meV when $w_1 = 90$ meV.

There are also regions in Fig. 4(a) in which the three-fold topological degeneracy of the ground states becomes poor and eventually collapses. On the left side of the FCI phase (with smaller $\theta$), we find a pronounced sensitivity of the energy spectrum to the lattice size. For $N = 6, N_1 \times N_2 = 3 \times 6$ and $N = 12, N_1 \times N_2 = 6 \times 6$, we observe a new kind of three-fold ground-state degeneracy in different momentum sectors from the $\nu = 1/3$ Laughlin FCIs. These momentum sectors are separated exactly by the moiré Dirac point momenta $\mathbf{K}_+^b$ and $\mathbf{K}_+^t$. For example, while the three Laughlin FCIs of $N = 12, N_1 \times N_2 = 6 \times 6$ all carry $(K_1, K_2) = (0, 0)$ [Fig. 5(a)], the three ground states at the parameter point $(\theta, P) = (0.95°, 20 \text{ meV})$ [labelled by the dot in Figs. 4(a)-4(c)] of the same system size are located in the $(K_1, K_2) = (0, 0), (2, 2)$ and $(4, 4)$ sectors [Fig. 6(c)], which are separated by momentum $\Delta \mathbf{K} = (2, 2) = (\mathbf{G}_1 + \mathbf{G}_2)/3 \sim \mathbf{K}_+^b$ and $\Delta \mathbf{K} = (4, 4) = 2(\mathbf{G}_1 + \mathbf{G}_2)/3 \sim \mathbf{K}_+^t$ (Fig. 1). Here $\sim$ means "equal to" up to a MBZ reciprocal lattice vector. The distribution of degenerate ground states over equally spaced momenta is a signal of charge density waves. To further confirm this, we compute the structure factor

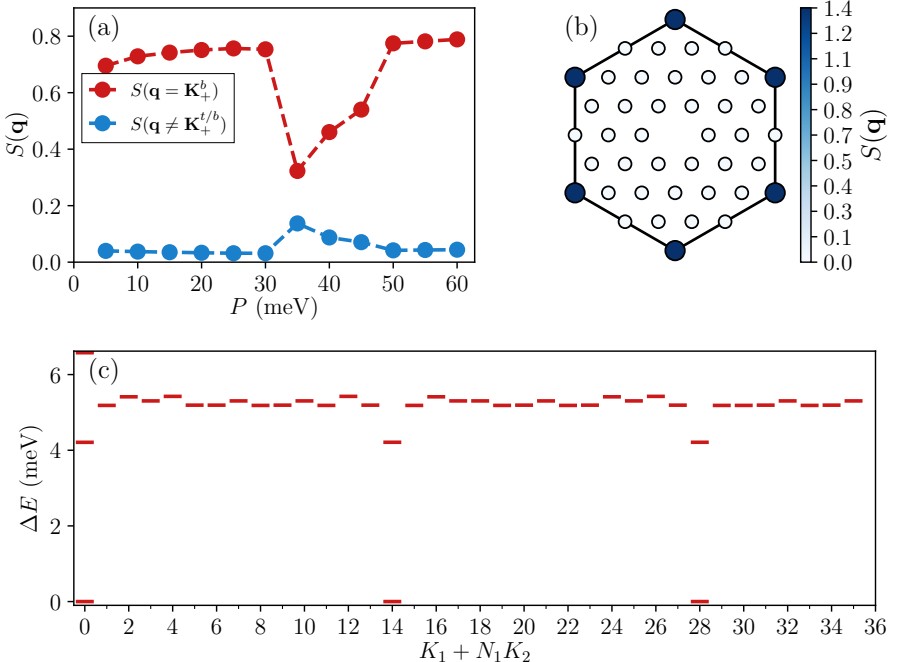

Figure 6: (a) The structure factor at $\theta = 0.95°$ as a function of $P$ for $N = 6$, $N_1 \times N_2 = 3 \times 6$. The values of $S(\mathbf{q})$ at $\mathbf{q} = \mathbf{K}_+^{t,b}$ are significantly larger than those at $\mathbf{q} \neq \mathbf{K}_+^{t,b}$. (b) Distribution of $S(\mathbf{q})$ in the MBZ at $(\theta, P) = (0.95°, 20 \text{ meV})$ for $N = 12$, $N_1 \times N_2 = 6 \times 6$. (c) The many-body energy spectrum at the same parameters with (b). The lowest three states are located at $(K_1, K_2) = (0,0), (2,2), (4,4)$, separated by $\Delta\mathbf{K} = (2,2)$ and $\Delta\mathbf{K} = (4,4)$.

$S(\mathbf{q})$ which can reveal the CDW order. We define $S(\mathbf{q})$ as

$$S(\mathbf{q}) = \frac{1}{N_1 N_2}\left(\langle \bar{\rho}(\mathbf{q})\bar{\rho}(-\mathbf{q})\rangle - N^2 \delta_{\mathbf{q},\mathbf{0}}\right), \tag{13}$$

where

$$\bar{\rho}(\mathbf{q}) = \sum_{\mathbf{k} \in \text{MBZ}} \langle u(\mathbf{k})|u(\mathbf{k}-\mathbf{q})\rangle c_{\mathbf{k}}^\dagger c_{\mathbf{k}-\mathbf{q}}. \tag{14}$$

Remarkably, we find pronounced peaks at the corners of the MBZ [Figs. 6(a) and 6(b)], revealing the underlying phase is the CDW with the order momentum $\mathbf{K}_+^{t,b}$. Various types of CDW phases have been identified in static TBG-hBN as competing phases against FCIs [21,67–69]. The $\mathbf{K}$-CDW states corresponds to a Wigner crystal, whose unit cell is tripled compared to the original moiré lattice [21]. On the other hand, we do not see the CDW degeneracy among the lowest states for $N = 8, N_1 \times N_2 = 4 \times 6$ and $N = 10, N_1 \times N_2 = 5 \times 6$. This is because the $\mathbf{K}_+^{t,b}$ points are absent in the MBZ of these finite lattices. Remember that the single-electron momentum $\mathbf{k}$ can only take $\mathbf{k} = \frac{m_1}{N_1}\mathbf{G}_1 + \frac{m_2}{N_2}\mathbf{G}_2$ for a finite periodic system. Therefore, both $N_1$ and $N_2$ must be divisible by three if $\mathbf{K}_+^{t,b}$ belong to this set of allowed $\mathbf{k}$. Otherwise, the $\mathbf{K}$-CDW cannot develop in the finite system.

Finally, we go to the region on the right side of the FCI phase (larger $\theta$), where we see neither the FCI topological degeneracy nor the CDW degeneracy in the low-energy spectra. To explore the nature of the ground states in this region, we switch from the picture of electrons to that of holes, i.e., changing $c(\mathbf{k})$ to $d^\dagger(\mathbf{k})$ in the Hamiltonian Eq. (11). Here $d^\dagger(\mathbf{k})$ creates a

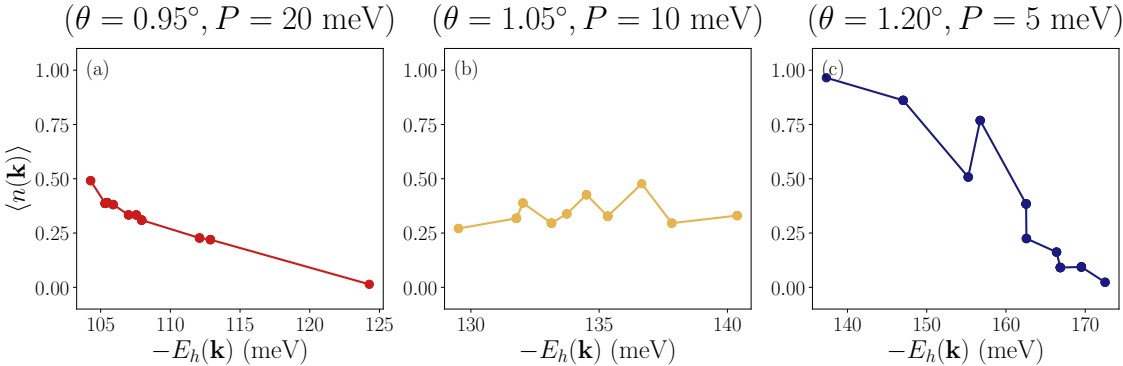

Figure 7: The ground-state occupation $\langle n(\mathbf{k})\rangle$ at momentum $\mathbf{k}$ as a function of $-E_h(\mathbf{k})$ for three representative parameter points. (a) and (b) correspond to the already identified CDW and FCI phase, respectively.

hole with momentum $\mathbf{k}$ in the Floquet valence band. As shown in Refs. [18, 70], this particle-hole transformation induces an effective hole dispersion

$$E_h(\mathbf{k}) \quad = \quad \sum_{\mathbf{k}'}(V_{\mathbf{k}'\mathbf{k}\mathbf{k}'\mathbf{k}} + V_{\mathbf{k}\mathbf{k}'\mathbf{k}\mathbf{k}'} - V_{\mathbf{k}\mathbf{k}'\mathbf{k}'\mathbf{k}} - V_{\mathbf{k}'\mathbf{k}\mathbf{k}\mathbf{k}'}) - E(\mathbf{k}) \tag{15}$$

in the dual Hamiltonian of holes. As $E_h(\mathbf{k})$ in general is non-constant, it breaks the particle-hole symmetry of Eq. (11). If such a dispersion dominates over the hole-hole interaction, the system can form a Fermi liquid-like state, that is, electrons (holes) tend to occupy $\mathbf{k}$ points with large (small) $E_h(\mathbf{k})$. This can be manifest by a sharp step in the electronic occupation number $\langle n(\mathbf{k})\rangle$ at some electron's "Fermi energy" $-E_h(\mathbf{k})$. In Fig. 7, we display the ground-state occupation $\langle n(\mathbf{k})\rangle$ at momentum $\mathbf{k}$ as a function of $-E_h(\mathbf{k})$ for the three representative parameter points [indicated by the dot, star, and triangle in Figs. 4(a)-4(c), respectively]. The $\langle n(\mathbf{k})\rangle$ curves show obviously distinct features in these three cases. In the FCI phase, the correlation between $\langle n(\mathbf{k})\rangle$ and $-E_h(\mathbf{k})$ is quite weak, being consistent with the expectation of $\langle n(\mathbf{k})\rangle \approx \nu$ for the FCIs [Fig. 7(b)]. The dependence of $\langle n(\mathbf{k})\rangle$ on $-E_h(\mathbf{k})$ becomes stronger in the CDW phase [Fig. 7(a)], as electrons tend to occupy $\mathbf{K}_+^{t,b}$ points. Eventually, for ground states at parameter points on the right side of the FCI phase, we find striking Fermi surface-like structures, that is, $\langle n(\mathbf{k})\rangle \approx 1$ for small $-E_h(\mathbf{k})$ and suddenly drops for larger $-E_h(\mathbf{k})$ [Fig. 7(c)]. This feature strongly suggests that the region on the right side of the FCI phase is a Fermi-liquid like phase.

Based on numerical results above, we present a tentative phase diagram for $w_1 = 90$ meV in Fig. 4(a). Because of the limited system sizes which we can reach by exact diagonalization, the phase boundaries are only roughly determined by the topological degeneracy of FCIs for 10 electrons, so they should not be thought as being precise.

## 4.2 $w_1 = 110$ **meV**

We have repeated the investigations in Sec. 4.1 for a stronger interlayer tunneling $w_1 = 110$ meV. We find the overall phase diagram [Figs. 4(d)-4(f)] is very similar to that for $w_1 = 90$ meV. However, the FCI phase exists at larger twist angles, which are still small but obviously deviate from the magic angle $\sim 1.05°$. This can be explained by the distribution of Berry curvature in the MBZ – we find that the twist angle has to be increased to achieve flatter Berry curvature when $w_1$ is strengthened from 90 meV to 110 meV. The best three-fold degeneracies of ten electrons appear at $\theta \approx 1.20° - 1.35°$ and $P \approx 5$ meV $- 35$ meV [Fig. 4(f)]. We demonstrate the energy spectrum and the PES at a representative parameter point [labelled by the star in

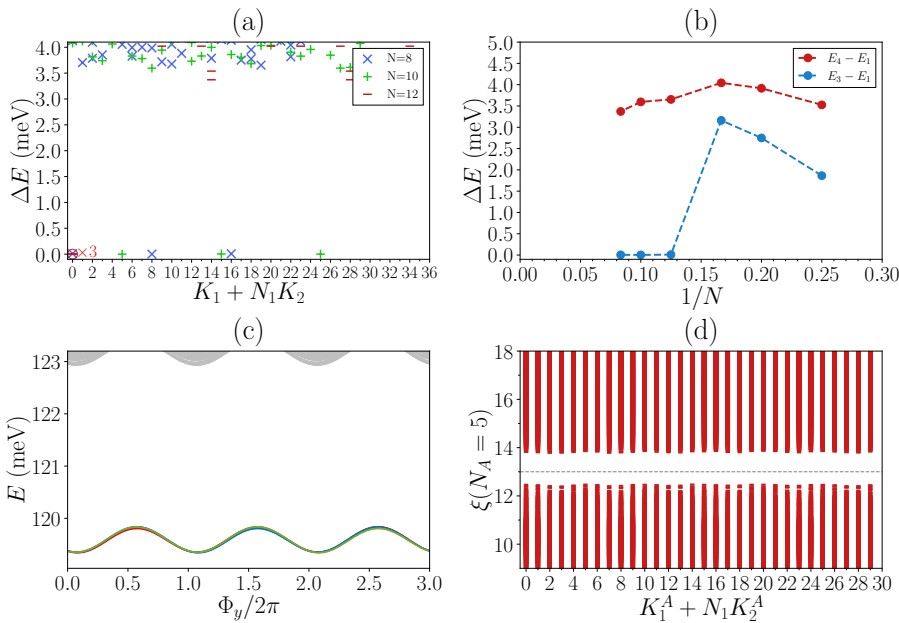

Figure 8: The $\nu = 1/3$ Laughlin FCIs in the Floquet valence band at $\theta = 1.27°$, $P = 10$ meV when $w_1 = 110$ meV. (a) The low-lying energy spectra for $N = 8, 10, 12$ electrons on the $N/2 \times 6$ lattice. (b) The finite-size scaling of the energy gap $E_4 - E_1$ and the ground-state splitting $E_3 - E_1$ for $N = 4, 5, 6, 8, 10, 12$ electrons. (c) The spectral flow for $N = 10$, $N_1 \times N_2 = 5 \times 6$, where $\Phi_y$ is the magnetic flux insertion in the $\mathbf{a}_2$ direction. (d) The particle entanglement spectrum for $N = 10$, $N_1 \times N_2 = 5 \times 6$ and $N_A = 5$, with 23256 levels below the entanglement gap (the dashed line).

Figs. 4(d)-4(f)] in this region (Fig. 8), where the energy gap protecting FCI ground states is even larger than that at $w_1 = 90$ meV. Meanwhile, the competition between FCI and CDW remains for $w_1 = 110$ meV. The signals of the CDW phase and the Fermi liquid-like phase, namely, the structure factor and the $\langle n(\mathbf{k}) \rangle - E_h(\mathbf{k})$ curve, are shown in Fig. 9 for two representative parameter points [labelled by the dot and triangle in Figs. 4(d)-4(f), respectively].

## 5 Discussion

In this work, we investigate the interaction effect in twisted bilayer graphene irradiated with monochromatic circularly polarized light. We work in the regime of high driving frequency. When the Floquet valence band obtained from the effective static Hamiltonian is partially occupied by electrons at $\nu = 1/3$ filling, we find compelling numerical evidence that Floquet fractional Chern insulators exist for a wide range of twist angle and driving strength, as characterized by the robust ground-state topological degeneracy and the counting of quasihole excitations. By calculating the structure factor and the electron occupation numbers, the intriguing interplay of Floquet FCIs with charge density waves and Fermi liquid-like states are also identified in a single phase diagram. Our results demonstrate the rich many-body physics in Floquet TBG. As the effective Hamiltonian in our setup is analogous to the static TBG-hBN system with hBN alignment on both top and bottom sides of TBG, the phase diagram obtained by us may also be helpful to the experimental realization of zero-field FCI in the static TBG-hBN system. Our results imply that the zero-field FCI could be stabilized in static TBG-hBN at twist angles larger than the magic value.

There are several interesting theoretical future directions following our present work. First,

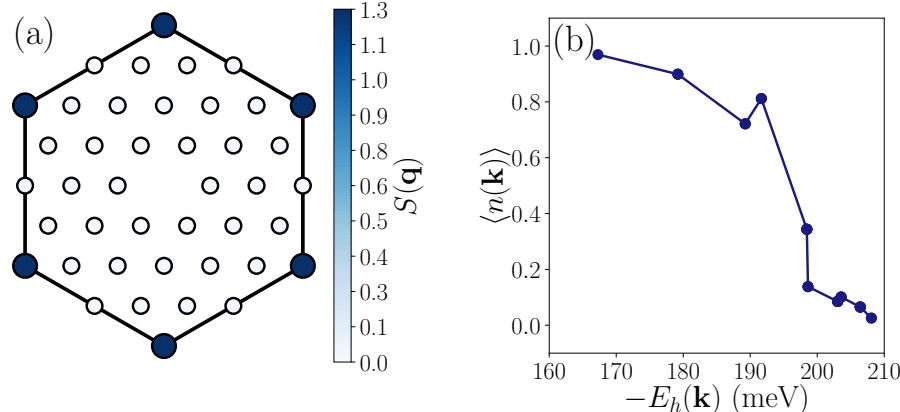

Figure 9: (a) Distribution of $S(\mathbf{q})$ in the MBZ for $N = 12$, $N_1 \times N_2 = 6 \times 6$ at $(\theta, P) = (1.20°, 55 \text{ meV})$. (b) The ground-state occupation $\langle n(\mathbf{k}) \rangle$ at momentum $\mathbf{k}$ as a function of $-E_h(\mathbf{k})$ for $N = 12$, $N_1 \times N_2 = 6 \times 6$ at $(\theta, P) = (1.50°, 10 \text{ meV})$.

it would be interesting to take into account more model parameters, such as the ratio $w_0/w_1$ and the dielectric constant, to explore the many-body phase diagram in a larger parameter space. Second, while we only consider $\nu = 1/3$ filling in this work, it remains unclear whether FCIs at other fillings, especially non-Abelian ones, can be stabilized in this Floquet system. Finally, inspired by the recent development in Floquet band structures of other moiré materials beyond TBG [40–43], it is natural to study the Floquet many-body physics thereof.

Considering that the Floquet Chern insulator has been realized in experiments of monolayer graphene [45], it is possible to observe it also in TBG driven by the light field, which is the first step towards realizing the Floquet FCIs predicted in our work. In fact, the parameters we choose here should be within experimentally realizable parameters. For example, both the driving frequency $\hbar\Omega = 1.5$ eV and the driving strength $\mathcal{E}_0 = 8$ MV/cm (corresponding to $P \approx 60$ meV) are accessible by current laser technology. However, obvious challenges still exist. For instance, the Floquet heating out of the long-lived prethermal regime will result a featureless infinite-temperature system. Moreover, one should keep the driving off-resonant, otherwise the direct absorption of photons by electrons makes the effective static Hamiltonian insufficient to capture the out-of-equilibrium properties (such as the transport) of the system [24].

# Acknowledgements

This project is supported by the National Key Research and Development Program of China through Grant No. 2020YFA0309200 and the National Natural Science Foundation of China through Grant No. 11974014. We thank the Tianhe-II platform at the National Supercomputer Center in Guangzhou for their allocation of CPU time.

# A Effective Floquet Hamiltonian

In this Appendix, we give a detailed derivation of the static effective Hamiltonian $H_{\text{eff}}$. Note that we do this derivation before the band projection. Within each layer of TBG, the Dirac

Hamiltonian of monolayer graphene under driving becomes

$$h(\mathbf{k}) \to h\left(\mathbf{k} + \frac{e}{\hbar}\mathbf{A}(t)\right) = -\hbar v_F \begin{pmatrix} 0 & k_- + \frac{eA_0}{\hbar}e^{i\Omega t} \\ k_+ + \frac{eA_0}{\hbar}e^{-i\Omega t} & 0 \end{pmatrix}, \tag{16}$$

where $k_\pm = k_x \pm i k_y$. The interlayer tunneling remains unchanged since we have neglected its transverse components. Then the only non-zero Fourier components $H_m = \frac{1}{T}\int_0^T H(t)e^{-im\Omega t}dt$ are

$$
\begin{aligned}
H_0 &= H_{\text{kin}} + H_{\text{int}}, \\
H_1 &= (-eA_0 v_F)\sum_{\mathbf{k}}\left(e^{i\theta/2}\psi_{tA}^\dagger(\mathbf{k})\psi_{tB}(\mathbf{k}) + e^{-i\theta/2}\psi_{bA}^\dagger(\mathbf{k})\psi_{bB}(\mathbf{k})\right), \\
H_{-1} &= H_1^\dagger,
\end{aligned}
\tag{17}
$$

where $H_{\text{kin}}$ is static TBG single-particle Hamiltonian defined in main text. Using the relation $\left[c_m^\dagger c_n, c_k^\dagger c_l\right] = \delta_{n,k}c_m^\dagger c_l - \delta_{m,l}c_k^\dagger c_n$, we have the first-order term of $H_{\text{eff}}$ as

$$H_{\text{eff}}^{(1)} = \frac{[H_1, H_{-1}]}{\hbar\Omega} = P\sum_{\mathbf{k}}\left(\psi_t^\dagger(\mathbf{k})\sigma_z\psi_t(\mathbf{k}) + \psi_b^\dagger(\mathbf{k})\sigma_z\psi_b(\mathbf{k})\right), \tag{18}$$

with $P = \frac{(eA_0 v_F)^2}{\hbar\Omega}$. The second-order term of $H_{\text{eff}}$ is

$$
\begin{aligned}
H_{\text{eff}}^{(2)} &= \frac{1}{2(\hbar\Omega)^2}[H_1, [H_0, H_{-1}]] + \text{H.c.} \\
&= \frac{1}{2(\hbar\Omega)^2}\left([H_1, [H_{\text{kin}}, H_{-1}]] + [H_1, [H_{\text{int}}, H_{-1}]]\right) + \text{H.c.}
\end{aligned}
\tag{19}
$$

A straightforward calculation gives

$$[H_1, [H_{\text{kin}}, H_{-1}]] + \text{H.c.} = 2(eA_0 v_F)^2\left[-H_{\text{kin}} + \sum_{\mathbf{k}}\sum_{j=1}^2\left(\psi_t^\dagger(\mathbf{k} - \mathbf{q}_0 + \mathbf{q}_j)\mathbb{W}_\theta\psi_b(\mathbf{k}) + \text{H.c.}\right)\right] \tag{20}$$

with

$$\mathbb{W}_\theta = w_0\begin{pmatrix} e^{i\theta} & 0 \\ 0 & e^{-i\theta} \end{pmatrix}. \tag{21}$$

The calculation of $[H_1, [H_{\text{int}}, H_{-1}]]$ is more tedious. We write the interaction Eq. (3) in the second-quantized form

$$H_{\text{int}} = \frac{1}{2}\sum_{\{\mathbf{k}_i\},\mathbf{q}}\sum_{\alpha,\beta}\delta'_{\mathbf{k}_1-\mathbf{k}_4,\mathbf{q}}\delta'_{\mathbf{k}_1+\mathbf{k}_2,\mathbf{k}_3+\mathbf{k}_4}V(\mathbf{q})\psi_\alpha^\dagger(\mathbf{k}_1)\psi_\beta^\dagger(\mathbf{k}_2)\psi_\beta(\mathbf{k}_3)\psi_\alpha(\mathbf{k}_4), \tag{22}$$

where $\alpha, \beta \in (tA, tB, bA, bB)$ are layer and sublattice indices. For fermionic creation and annihilation operators, we have

$$
\begin{aligned}
&\left[c_\alpha^\dagger(\mathbf{k}_1)c_\beta^\dagger(\mathbf{k}_2)c_\gamma(\mathbf{k}_3)c_\delta(\mathbf{k}_4), \sum_{\mathbf{k}}f(\mathbf{k})c_\mu^\dagger(\mathbf{k})c_\nu(\mathbf{k})\right] \\
=\ & \delta_{\mu,\delta}f(\mathbf{k}_3)c_\alpha^\dagger(\mathbf{k}_1)c_\beta^\dagger(\mathbf{k}_2)c_\gamma(\mathbf{k}_3)c_\nu(\mathbf{k}_4) + \delta_{\mu,\gamma}f(\mathbf{k}_4)c_\alpha^\dagger(\mathbf{k}_1)c_\beta^\dagger(\mathbf{k}_2)c_\nu(\mathbf{k}_3)c_\delta(\mathbf{k}_4) \\
-\ & \delta_{\nu,\alpha}f(\mathbf{k}_1)c_\mu^\dagger(\mathbf{k}_1)c_\beta^\dagger(\mathbf{k}_2)c_\gamma(\mathbf{k}_3)c_\delta(\mathbf{k}_4) - \delta_{\nu,\beta}f(\mathbf{k}_2)c_\alpha^\dagger(\mathbf{k}_1)c_\mu^\dagger(\mathbf{k}_2)c_\gamma(\mathbf{k}_3)c_\delta(\mathbf{k}_4)
\end{aligned}
\tag{23}
$$

for an arbitrary function $f(\mathbf{k})$. We then apply Eq. (23) to $[H_{\text{int}}, H_{-1}]$, and get

$$
\begin{aligned}
[H_{\text{int}}, H_{-1}] &= (-eA_0 v_F) \sum_{\{\mathbf{k}_i\}, \mathbf{q}} \sum_{\alpha, \beta} \delta'_{\mathbf{k}_1 - \mathbf{k}_4, \mathbf{q}} \delta'_{\mathbf{k}_1 + \mathbf{k}_2, \mathbf{k}_3 + \mathbf{k}_4} V(\mathbf{q}) \\
&\times \{e^{-i\theta/2} \psi^\dagger_{tB}(\mathbf{k}_1) \psi^\dagger_\alpha(\mathbf{k}_2) \psi_\alpha(\mathbf{k}_3) \psi_{tA}(\mathbf{k}_4) + e^{-i\theta/2} \psi^\dagger_\alpha(\mathbf{k}_1) \psi^\dagger_{tB}(\mathbf{k}_2) \psi_{tA}(\mathbf{k}_3) \psi_\alpha(\mathbf{k}_4) \\
&\quad - e^{-i\theta/2} \psi^\dagger_{tB}(\mathbf{k}_1) \psi^\dagger_\alpha(\mathbf{k}_2) \psi_\alpha(\mathbf{k}_3) \psi_{tA}(\mathbf{k}_4) - e^{-i\theta/2} \psi^\dagger_\alpha(\mathbf{k}_1) \psi^\dagger_{tB}(\mathbf{k}_2) \psi_{tA}(\mathbf{k}_3) \psi_\alpha(\mathbf{k}_4) \\
&\quad + e^{i\theta/2} \psi^\dagger_{bB}(\mathbf{k}_1) \psi^\dagger_\alpha(\mathbf{k}_2) \psi_\alpha(\mathbf{k}_3) \psi_{bA}(\mathbf{k}_4) + e^{i\theta/2} \psi^\dagger_\alpha(\mathbf{k}_1) \psi^\dagger_{bB}(\mathbf{k}_2) \psi_{bA}(\mathbf{k}_3) \psi_\alpha(\mathbf{k}_4) \\
&\quad - e^{i\theta/2} \psi^\dagger_{bB}(\mathbf{k}_1) \psi^\dagger_\alpha(\mathbf{k}_2) \psi_\alpha(\mathbf{k}_3) \psi_{bA}(\mathbf{k}_4) - e^{i\theta/2} \psi^\dagger_\alpha(\mathbf{k}_1) \psi^\dagger_{bB}(\mathbf{k}_2) \psi_{bA}(\mathbf{k}_3) \psi_\alpha(\mathbf{k}_4)\} \\
&= 0. \tag{24}
\end{aligned}
$$

Therefore, only $H_{\text{kin}}$ contributes to $H^{(2)}_{\text{eff}}$ in our model, making $H^{(2)}_{\text{eff}}$ a non-interacting part in $H_{\text{eff}}$. This is different from the result in Ref. [47], whose authors obtained effective interactions in $H^{(2)}_{\text{eff}}$ for another Floquet lattice model. The reason for this difference is that, in our model the factor in front of $\psi^\dagger(\mathbf{k})\psi(\mathbf{k})$ in $H^{(1)}_{\text{eff}}$ does not depend on $\mathbf{k}$, and $\alpha$ and $\beta$ in the interaction can take the same value in Eq. (22). On the contrary, in the model of Ref. [47] the corresponding prefactor in $H^{(1)}_{\text{eff}}$ does depend on $\mathbf{k}$ and $\alpha$ and $\beta$ in the interaction cannot take the same value. Hence, $[H_{\text{int}}, H_{-1}]$ is different between our model and the model in Ref. [47].

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
