# Peer review of "Floquet Fractional Chern Insulators and Competing Phases in Twisted Bilayer Graphene"

_SciPost Physics_

## Round 1 · Referee Report · Titus Neupert (Referee 1) · 2022-11-18

Strengths

1 Authors make effort to take realistic parameters of the system for their calculation.
2 Comprehensive explanation of the derivation of the interacting Floquet Hamiltonian.
3 Logical presentation that is easy to digest.
4 Clean results.

Weaknesses

1 An interacting Floquet Hamiltonian is treated as if it was a static Hamiltonian to determine the many-body ground state. I am not sure how well this procedure is justified.

Report

In the light of the points mentioned unter Strenghts/Weaknesses, I consider this a nice and timely piece of work that deserves publication in SciPost. Nevertheless, there are two points that need clarification in my view:
1 How is the assumption of valley and spin polarization justified? The authors assume this from the outset but do not comment on this.
2 If I understand correctly, Fig 3 is not the result of an interacting calculation. Why is it then not presented as a nice phase diagram like Fig. 2, but only so few points are calculated?

Requested changes

Those that follow under the points mentioned in the report.

---

## Round 1 · Referee Report · Anonymous (Referee 2) · 2023-1-11

Strengths

1- timely topic, as experimental teams are trying to achieve FCIs in the absence of a magnetic field in TBG 2- driving field is a convenient tuning knob: it appears to play the same role as alignment to hBN in other works. Not only does it circumvent the difficulty of aligning hBN and TBG, the driving field can be adjusted continuously to provide an extra, controllable parameter axis. 3- state-of-the-art numerics: strong numerical analysis of the FCI and competing phases 4- well-written, clear paper

Weaknesses

1- only little qualitative understanding of phase diagram through analysis of the FCI single-particle markers (deviation from uniform Berry curvature, deviation from saturation of trace inequality, bandwidth). 2- no discussion of the role of explicit time-reversal-symmetry breaking by the driving field. 3- spin and valley polarization are assumed, and the phase diagram could be more complex if this assumption is relaxed.

Report

The authors study magic angle twisted bilayer graphene (TBG) subject to a drive of circularly polarized light as a platform for the realization of fractional Chern insulators (FCIs), i.e. the fractional quantum Hall effect in the absence of a static magnetic field. They use Floquet theory to obtain an effective static Hamiltonian under the drive of frequency Omega. Up to second order in 1/Omega, they find that the drive does not affect the interaction term of the Hamiltonian (screened Coulomb interaction), while its main effect on the kinetic part of the Hamiltonian is the addition of a staggered sublattice potential of strength P akin to the (static) effect of alignment of graphene to the hexagonal boron nitride (hBN) substrate. A similar observation had been reported in previous Floquet treatment of TBG up to first order; the current manuscript clarifies that the second order term does significantly affect the Floquet Hamiltonian (no additional the interaction term, small - 4% - kinetic term). After deriving the Floquet Hamiltonian, the authors numerically study its many-body phase diagram at filling fraction nu = 1/3, as a function of the twist angle theta, and staggered potential P. They assume full valley and spin polarization, and project the interaction to a single band. The nature of the many-body ground state is carefully analyzed base on particle entanglement spectroscopy, flux insertion, static structure factor, and occupation of single-particle states, with appropriate finite-size extrapolation. The authors find three different phases: a FCI, a charge density wave (CDW), and a Fermi liquid. These different phases are the same ones already identified in other FCI models, especially TBG models in the absence of a drive.

I find the motivation for the present study to be relevant and timely, and the paper is clearly written, with convincing numerical evidence. Yet, some key elements are missing, in my opinion, in order to fully answer the questions the authors set out to address. If these elements were provided satisfyingly, I would recommend publication.

First, unlike hBN alignment, the circularly polarized drive explicitly breaks time-reversal symmetry (TRS). Therefore, I would expect some differences between the band structures of TBG + staggered AB-potential and TBG + staggered AB-potential + TRS breaking. Can the authors please clarify? This question has likely been explored in previous references on the floquet theory of TBG, but it would be relevant to make it explicit in the context of the many-body phase diagram. This is especially important because it could potentially affect valley polarization (assumed in the paper), which emerges spontaneously from strong interactions at some fillings in the TRS case.

Second, the authors have provided only small hints to microscopically explain the many-body phase diagram. As the authors are well aware, there are a few single-particle markers that can predict the emergence of a FCI in a Chern band: small enough deviation from a homogeneous Berry curvature, small enough deviation from the saturation of the trace inequality of the quantum metric tensor, and small enough bandwidth (the bandwidth is provided, but not the others). I think it would be helpful to compare the maps of these markers with the obtained phase diagram, to provide a more systematic understanding of their many-body results.

Finally, my last comments concern the FCI many-body gap. The authors have provided a value of around 20 Kelvins. What Coulomb interaction strength U was used to obtain this number? The experimental U may not be known to a great precision, so it may be more useful to express the energy spectrum and many-body gap in units of U. I am also surprised by this value, which is ten times larger than the value obtained in other references of FCIs in TBG (with no drive), using the value U ~ 20 meV. The discrepancy may be explained by the U value used by the authors, if not it would be important to understand its origin. Additionally, do Floquet FCIs require larger gaps (compared to intrinsic FCIS) in order to be observed experimentally (due to heating processes for example)? The authors state that the estimated gap is an order of magnitude larger than the ones that can be observed experimentally, so it would be good to clarify if this typical estimate of what is detectable takes into account the non-equilibrium nature of the system.

Requested changes

As indicated in the report.

---

## Editorial Decision

resubmitted